# Genomic and phenotypic diversity of *Enterococcus faecalis* isolated from endophthalmitis

**Gayatri Shankar Chilambi[1], Hayley R. Nordstrom[1], Daniel R. Evans[1], Regis P. Kowalski[2], Deepinder K. Dhaliwal[2], Vishal Jhanji[2], Robert M. Q. Shanks[2], Daria Van Tyne[1]***

1 Division of Infectious Diseases, University of Pittsburgh School of Medicine, Pittsburgh, Pennsylvania, United States of America, 2 The Charles T. Campbell Ophthalmic Microbiology Laboratory, UPMC Eye Center, Ophthalmology and Visual Sciences Research Center, The Eye and Ear Institute, Department of Ophthalmology, University of Pittsburgh School of Medicine, Pittsburgh, Pennsylvania, United States of America

* vantyne@pitt.edu

**Data Availability Statement:** Illumina read data for isolates newly sequenced in this study have been submitted to NCBI under BioProject PRJNA649986. All other relevant data are within the paper and its Supporting Information files.

## Abstract

*Enterococcus faecalis* are hospital-associated opportunistic pathogens and also causative agents of post-operative endophthalmitis. Patients with enterococcal endophthalmitis often have poor visual outcomes, despite appropriate antibiotic therapy. Here we investigated the genomic and phenotypic characteristics of *E. faecalis* isolates collected from 13 patients treated at the University of Pittsburgh Medical Center Eye Center over 19 years. Comparative genomic analysis indicated that patients were infected with *E. faecalis* belonging to diverse multi-locus sequence types (STs) and resembled *E. faecalis* sampled from clinical, commensal, and environmental sources. We identified known *E. faecalis* virulence factors and antibiotic resistance genes in each genome, including genes conferring resistance to aminoglycosides, erythromycin, and tetracyclines. We assessed all isolates for their cytolysin production, biofilm formation, and antibiotic susceptibility, and observed phenotypic differences between isolates. Fluoroquinolone and cephalosporin susceptibilities were particularly variable between isolates, as were biofilm formation and cytolysin production. In addition, we found evidence of *E. faecalis* adaptation during recurrent endophthalmitis by identifying genetic variants that arose in sequential isolates sampled over eight months from the same patient. We identified a mutation in the DNA mismatch repair gene *mutS* that was associated with an increased rate of spontaneous mutation in the final isolate from the patient. Overall this study documents the genomic and phenotypic variability among *E. faecalis* causing endophthalmitis, as well as possible adaptive mechanisms underlying bacterial persistence during recurrent ocular infection.

## Introduction

Enterococci are Gram-positive bacteria that inhabit the gastrointestinal tract of humans and other land animals [1]. Enterococci also cause a wide variety of infections, particularly in

**Funding:** This study was funded by PHS grant EY028222 to DVT and by the University of Pittsburgh Department of Medicine. The funders had no role in study design, data collection and analysis, decision to publish, or preparation of the manuscript.

**Competing interests:** The authors have declared that no competing interests exist.

immunocompromised hosts. Two members of the *Enterococcus* genus, *E. faecium* and *E. faecalis*, account for nearly 75% of all enterococcal infections [2]. Multidrug-resistant *E. faecalis* are leading causes of healthcare-associated infections, including bacteremia, endocarditis, and urinary tract, intra-abdominal, surgical site, and device-associated infections [3]. The prevalence of enterococci in the modern hospital environment is attributed to their intrinsic resistance to commonly used antibiotics, their ability to survive under harsh and stressful conditions, their propensity to acquire mobile genetic elements carrying drug resistance and other pathogenicity-enhancing genes, and their ability to form biofilms on indwelling medical devices such as catheters and intraocular lenses [4–6].

Endophthalmitis is characterized by inflammation of the interior of the eye. It is often caused by infection, either as a consequence of intraocular surgery (post-operative endophthalmitis), penetrating injury (post-traumatic endophthalmitis), or spread of bacteria from a distant site of infection to the eye (endogenous endophthalmitis) [7]. Most cases of bacterial endophthalmitis are caused by coagulase-negative staphylococci (CoNS), *Staphylococcus aureus* and viridans group streptococci [8]. *E. faecalis* are infrequently isolated from post-operative endophthalmitis, but patients with post-operative enterococcal endophthalmitis have worse clinical outcomes compared with similar infections caused by CoNS and *S. aureus* [9]. The increasing occurrence of *E. faecalis* as a causative organism of post-operative endophthalmitis has been reported across the world [10–12].

The purpose of this study was to characterize the genomic and phenotypic features of 15 *E. faecalis* isolates from post-operative endophthalmitis collected from 13 patients treated at the University of Pittsburgh Medical Center (UPMC) Eye Center. In addition to assessing the genomic diversity of these isolates, we also analyzed their plasmid, antibiotic resistance gene, CRISPR-Cas, prophage and virulence gene profiles. We measured the susceptibility of each isolate to antibiotics commonly used to treat enterococcal endophthalmitis, as well as their ability to produce the cytolysin toxin and form biofilms. Where possible, we made connections between these phenotypes and bacterial genotypes. Finally, we identified factors contributing to the persistent colonization of *E. faecalis* in one patient with recurrent endophthalmitis [13].

## Methods

### Collection of isolates

Bacterial isolates were collected from patients seeking treatment at the UPMC Eye Center between 1993 and 2012. The clinical specimens were processed in the Charles T. Campbell Ophthalmic Microbiology Laboratory, University of Pittsburgh Medical Center, Pittsburgh, PA. In general, samples of aqueous and vitreous humor of patients with endophthalmitis were cultured for pathogens. The specimens were collected directly by tapping eye chambers using tuberculin syringes and needles. An approximate volume of 0.2–0.3 mL of intraocular fluid was collected from each patient. Intraocular fluid was placed on two glass slides for rapid visualization of microorganisms using Gram and Giemsa staining. The remaining fluid was dispersed on various isolation medium, including trypticase soy agar supplemented with 5% sheep blood, chocolate agar, enriched thioglycolate broth, a chocolate agar plate incubated in an anaerobic bag, and Sabouraud dextrose medium with gentamicin for fungal detection. Most aerobic plates were incubated at 37˚C in a $CO_2$ atmosphere. The Sabouraud dextrose medium was incubated in a 30˚C air incubator. *Enterococcus faecalis* grew readily on all culture media. The isolates were Gram-positive cocci (coccoidal), catalase-negative, pyrrolidonyl arylamidase-positive, and colonies were greyish in nature. After laboratory isolation, *E. faecalis* isolates were patient de-identified, and were stored at -80˚C in broth medium containing 15% glycerol. Isolates were saved as part of a clinical tissue bank used for testing and antimicrobial

susceptibility validation. The study was approved by the University of Pittsburgh Institutional Review Board under protocol number STUDY19110081.

## Whole genome sequencing and analysis

Genomic DNA from 15 post-operative endopthalmitis isolates was extracted using a DNeasy Blood and Tissue Kit (Qiagen, Germantown, MD) from 1mL bacterial cultures grown in Brain Heart Infusion (BHI) media. Next-generation sequencing libraries were prepared with a Nextera XT kit (Illumina, San Diego, CA), and libraries were sequenced on an Illumina MiSeq using 300bp paired-end reads. Genomes were assembled with SPAdes v3.13.0 [14], annotated with prokka [15], and were compared to one another with Roary [16]. Genome assemblies of additional *E. faecalis* from endophthalmitis (n = 2) and other sources (n = 49) were downloaded from the NCBI database (see S1 and S2 Tables). Core genome phylogenetic trees were generated using RAxML with the GTRCAT substitution model and 100 iterations [17]. Sequence types, antimicrobial resistance genes, virulence factors, and plasmid replicons were identified using online tools provided by the Center for Genomic Epidemiology (https://cge. cbs.dtu.dk/). CRISPR-cas loci and prophage sequences were identified using CRISPRCasFinder [18] and PHASTER (PHAge Search Tool Enhanced Release), respectively [19]. Variants were identified in serial isolates from the same patient using CLC Genomics Workbench v11.0.1 (Qiagen, Germantown, MD), using a read depth cut-off of 10 reads and a variant frequency cut-off of >90%. Illumina read data for isolates newly sequenced in this study have been submitted to NCBI under BioProject PRJNA649986, with accession numbers listed in S1 Table.

## Antimicrobial susceptibility testing

Susceptibility testing to determine the minimum inhibitory concentration (MIC) of amikacin, benzalkonium chloride, ceftazidime, moxifloxacin, ofloxacin, povidone-iodine and vancomycin was carried out by broth microdilution in Mueller Hinton Broth (MHB) [20]. Briefly, overnight cultures of *E. faecalis* grown in MHB were diluted to an $OD_{600}$ of 0.1, and were further diluted 1:1000 into fresh MHB. 100μL of this culture was then transferred to 96-well plates containing 100μL of MHB with serial two-fold dilutions of each antimicrobial, yielding approximately $10^5$ bacteria per well. Plates were incubated for 24 hours at 37˚C under static conditions, and growth in each well was analyzed by both visual inspection and by $OD_{600}$ measurement using a Synergy H1 microplate reader (Biotek, Winooski, VT). Assays were conducted in triplicate and resistance was assessed according to criteria established by the Clinical and Laboratory Standards Institute [21].

## Biofilm assay

Microtiter plate-based biofilm assays were performed as previously described [22]. Briefly, overnight cultures of each isolate were diluted 100-fold into BHI broth supplemented with 0.25% glucose. 200μL of each culture was plated into eight replicate wells of a 96-well untreated polystyrene microtiter plate, and plates were incubated for 24 hours at 37˚C under static conditions. Planktonic cells were removed and plates were washed three times with 250μL 1xPBS, then wells were stained with 200μL 0.1% crystal violet (CV) in water. After incubation for 30 minutes at 4˚C, stained wells were washed twice with 250μL 1xPBS to remove excess stain. Plates were dried and then 250μL of 4:1 ethanol:acetone was added to each well to solubilize CV-stained biofilms. After incubation for 45 minutes at room temperature, the absorbance in each well was measured at 550nm using a Synergy H1 microplate reader (Biotek, Winooski, VT). Negative control (NC) wells contained 200μL of BHI broth supplemented with 0.25%

glucose and no bacteria. To compare biofilm growth with stationary phase cell density, the assay was repeated as described above, however after 24 hours of incubation at 37˚C under static conditions each well was resuspended and the optical density at 600nm was recorded.

## Cytolysin activity

The beta-hemolytic activity of the 15 UPMC endophthalmitis isolates was evaluated by streaking each isolate on BHI agar supplemented with 5% defibrinated horse blood. Plates were incubated for 24 hours at 37˚C. A clear zone around the streaked bacteria was recorded as positive for beta-hemolysis.

## Quantification of spontaneous mutation rate

The rates of spontaneous mutation in the E616, E623, and E633 isolates were determined by a previously described protocol [23]. Briefly, 50μL of overnight bacterial culture grown in BHI media was plated onto BHI agar plates, each containing 50μg/mL of rifampin. The initial cell concentration was determined by plating serial dilutions of the overnight culture onto agar with no antibiotic. Colonies were counted after overnight incubation at 37˚C. The mutation frequency (μ) was determined by dividing the number of rifampin-resistant colonies on each plate by the initial inoculum.

## Statistics

Differences in antimicrobial resistance gene content, biofilm formation, and spontaneous mutation rates were assessed with a two-tailed t-test.

## Results

### Clinical presentation and management of patients with enterococcal endophthalmitis, and identification of *E. faecalis*

All 13 patients in the study were examined and treated by the Retina Service at the UPMC Eye Center. The mean age of the patients was 92.5 years (range 83–99 years). All patients presented with acute endophthalmitis. The presenting visual acuity was hand motion or less in all patients. Bacterial isolates were collected as part of routine clinical care. Vitreous and aqueous humor samples were collected during surgery and were sent to the Charles T. Campbell Ophthalmic Microbiology Laboratory for bacterial isolation. Clinical management was in the form of vitrectomy and injection of intravitreal antibiotics. The final visual acuity was hand motion or less in all patients.

### Genomic features of *E. faecalis* from endophthalmitis

A total of 15 *E. faecalis* isolates collected from 13 endophthalmitis patients treated at the UPMC Eye Center were available for genome sequencing and analysis. A search of the National Center for Biotechnology Information (NCBI) database identified two additional genomes isolated from patients with endophthalmitis in Denmark (Genbank accessions GCA_900205805.1 and GCA_900205785.1) [24], resulting in 17 genomes total. All isolates were sequenced on the Illumina platform, and the data were used to construct a core genome phylogeny (Fig 1). *In silico* multi-locus sequence typing (MLST) of all 17 genomes showed that 11 distinct sequence types (STs) were identified. No single ST was found to dominate; however, multiple isolates belonging to ST2, ST40, ST64, and ST122 were observed. In the case of the ST122 isolates, all three isolates were collected from a single patient with recurrent endophthalmitis [13]. To compare the 17 *E. faecalis* endophthalmitis genomes with other non-

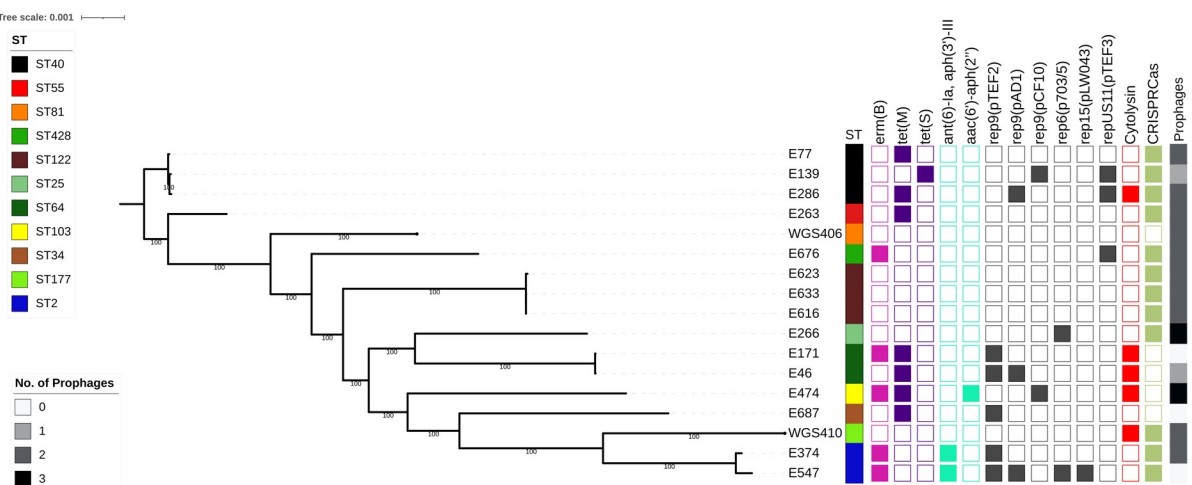

**Fig 1. Core genome phylogeny of *E. faecalis* endophthalmitis isolates.** Single-copy core genome phylogeny of 15 UPMC Campbell Lab isolates plus two additional genomes from NCBI (WGS406 and WGS410), all isolated from endophthalmitis. The RAxML tree is built from single nucleotide polymorphisms (SNPs) in 2024 single copy core genes identified with Roary. Tips are annotated with isolate name, multi-locus sequence type (ST), drug resistance-associated genes, plasmid *rep* genes, cytolysin operon presence, CRISPR-cas loci presence, and prophage abundance in each genome. CRISR-cas-positive isolates had Class 1 and/or Class 2 Cas proteins associated with their CRISPR loci. Intact or questionable prophages were identified with PHASTER. *erm(B)* = erythromycin resistance, *tet(M)* and *tet(S)* = tetracycline resistance, *ant(6)-Ia, aph(3′)-III* and *aac(6')-aph(2″)* = aminoglycoside resistance.

ophthalmic isolates, we also constructed a phylogeny including 49 diverse *E. faecalis* genomes sampled from clinical, commensal, and environmental sources [25] (S1 Fig). The endophthalmitis isolates were dispersed throughout the phylogenetic tree, confirming that no single genetic lineage was dominant among the ocular *E. faecalis* that we sampled.

We screened all 17 *E. faecalis* endophthalmitis genomes for acquired antimicrobial resistance genes using the ResFinder tool available through the Center for Genomic Epidemiology (CGE) [26] (Fig 1). All isolates were predicted to be intrinsically resistant to lincosamides due to the presence of *lsa(A)*, a core gene found in all *E. faecalis*. Additionally, five isolates carried the erythromycin resistance gene *erm(B)*, and eight isolates possessed either *tet(M)* or *tet(S)* tetracycline resistance genes. Finally, aminoglycoside resistance-conferring genes *ant(6)-Ia* and *aph(3′)-III* were identified in the two ST2 isolates, and *aac(6')-aph(2″)* was observed in the single ST103 isolate genome.

Next we screened all genomes for plasmid *rep* genes using PlasmidFinder [27], and identified six different *rep* genes that together were detected in ten (58.8%) isolate genomes (Fig 1). Among these, six isolates had only a single *rep* gene identified, while the remaining four had two or more *rep* genes. The E547 genome, belonging to ST2, encoded four different *rep* genes; this isolate also carried erythromycin and aminoglycoside resistance genes, which are often plasmid-encoded. While we did not observe any correlation between the *rep* families and the STs sampled, *rep* genes belonging to the *rep9* family were detected in eight isolate genomes, making it the dominant *rep* family observed in this study.

We then assessed the Clustered Regularly Interspaced Short Palindromic Repeats (CRISPR) and their associated Cas loci, as well as the prophage content, of the 17 *E. faecalis* endophthalmitis genomes. Genomes were mined using CRISPRCasFinder [28] and PHASTER [19]. We observed that all the isolates in our study had identifiable CRISPR sequences, however E46, E139, E171, E474, and WGS406 encoded orphan CRISPR sequences without any associated Cas proteins (S3 Table). CRISPR sequences without adjacent Cas proteins have been previously observed in *E. faecalis* [29, 30]. The remaining 12 isolates did have Class 1 and/or Class 2

Cas proteins associated with their CRISPR loci (S3 Table, Fig 1). The five genomes lacking an intact CRISPR-Cas locus had more acquired antimicrobial resistance genes on average compared to CRISPR-Cas-positive genomes, but the difference was not statistically significant (mean 1.6 vs. 0.75 genes per genome, $P = 0.09$ by two-tailed t-test). Prophage analysis with PHASTER [19] detected intact or questionable prophages in 14/17 (82%) of the enterococcal genomes in this study. Prophages varied in size from 20–58.9Kb, and showed homology to previously described *Enterococcus* and other Gram-positive phages (S4 Table). We did not observe any correlation between CRISRP-Cas loci and the number of plasmid *rep* genes or prophages identified in the study isolate genomes.

## Cytolysin activity

The *E. faecalis* cytolysin is a pore-forming toxin that lyses both bacterial and eukaryotic cells in response to quorum signals [31–34]. We detected the cytolysin operon in 5/17 (29%) endophthalmitis isolate genomes (Fig 1). To confirm cytolysin operon activity, we tested the beta-hemolytic capacity of the 15 UPMC isolates on agar plates containing 5% horse blood. As predicted from their genomes, E46 (ST64), E171 (ST64), E286 (ST40) and E474 (ST103) showed beta-hemolysis when grown in the presence of horse blood [35]. We were unable to test for cytolysin activity in WGS406 because we only had access to the genome sequence of this strain.

## Virulence-associated gene profiles and biofilm formation

We evaluated all 17 *E. faecalis* genomes for the presence of virulence-associated genes and operons using the VirulenceFinder database [36], along with manual searches for putative virulence gene sequences that have been previously described in *E. faecalis* [37] (S5 Table). We found that the fibronectin-binding proteins *efbA* and *efaAfs*, endocarditis and biofilm-associated pili *ebpABC*, sortase A (*srtA*), adhesin to collagen of *E. faecalis* (*ace*), general stress proteins *gls24 and glsB*, membrane metalloprotease associated with endocarditis (*eep*), thiol peroxidase for oxidative stress resistance (*tpx*), and sex pheromones *cCF10*, *cOB1*, *cad* and *camE* were present in the genomes of all isolates, with variable nucleotide identities (97–100% versus the reference sequence). Other virulence-associated genes were variably present, including the quorum-sensing gene *fsrB*, gelatinase *gelE*, serine protease *sprE*, hyaluronidase *hylAB*, and several genes often found within the *E. faecalis* pathogenicity island (PAI, S5 Table).

Biofilm formation is a well described pathogenicity-enhancing feature of many bacteria, including *E. faecalis* [38]. We measured the *in vitro* biofilm forming capacity of all 15 UPMC isolates using the crystal violet staining method [22], and observed differences in biofilm formation between isolates (Fig 2). These differences were not due to differences in bacterial growth rate, as there was no correlation between stationary phase optical density and biofilm formation among the tested isolates (S2 Fig). Isolate E286 (ST40) formed the most biofilm of any isolate tested, in contrast with the other two ST40 isolates, which formed only moderate biofilms ($P<0.0001$). Similarly, E171 (ST64) formed more biofilm than E46, another ST64 isolate ($P<0.0001$). Both ST2 isolates showed similar, moderate levels of biofilm formation, and the three ST122 isolates from the same patient were similar to one another in their ability to form biofilms, and were among the weakest biofilm formers of all the isolates tested.

To identify candidate genes that might contribute to enhanced biofilm formation in E286 (ST40) and E171 (ST64), we examined the sequences of biofilm-associated genes between these isolates and other isolates of the same STs that formed weaker biofilms (S5 Table). The sequence of the collagen adhesin *ace* was intact in E286, but appeared to be disrupted in the other two ST40 isolates (S3 Fig). Separately, the genome of the E46 isolate, which formed less

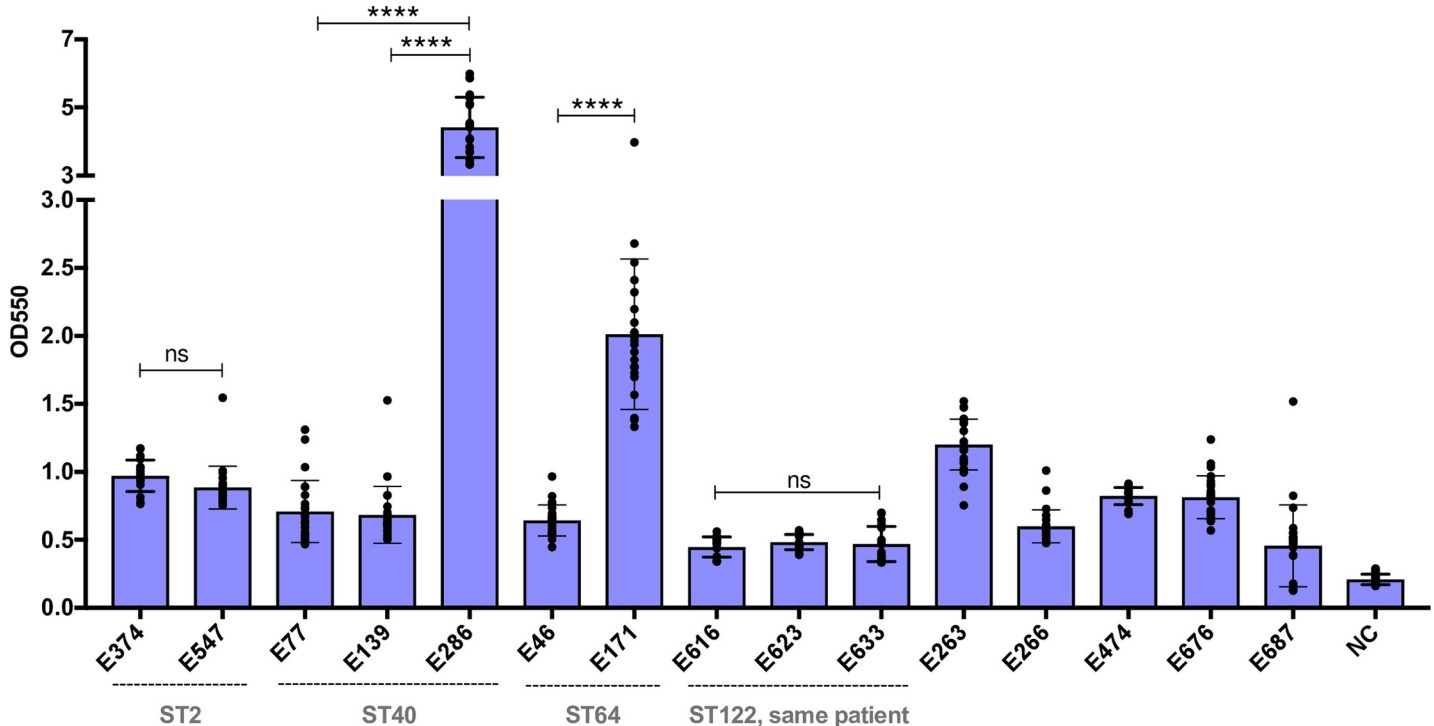

**Fig 2. Variable biofilm production among endophthalmitis isolates.** *In vitro* biofilm production of 15 *E. faecalis* endophthalmitis isolates. Biofilm formation was measured as the optical density (OD) at 550nm using a standard crystal violet-based assay. Isolates are arranged according to their sequence type (ST), and STs with more than one isolate are labeled. Bars shown mean crystal violet absorbance values, and error bars show standard deviation of triplicate experiments, each with eight technical replicates. NC = negative control; ****$P<0.0001$ by two-tailed t test; ns = not significant.

biofilm than the other ST64 isolate (E171), also had a disruption in the *ace* gene (S3 Fig). We also examined differences in gene content among the ST40 and ST64 isolates (S6 Table). Genes that were differentially present in E286 compared to E77 and E139 included prophage and plasmid-associated genes, while among the genes that were present in E171 but absent in E46 was an Asa373-like protein. This protein contains a BspA domain which has been previously associated with adhesion in Group B streptococci [39], and which has been shown to cause cell aggregation in *E. faecalis* [40].

## Antimicrobial susceptibility phenotypes and correlation with genotypes

We investigated the antimicrobial susceptibilities of the 15 endophthalmitis isolates we collected by determining the minimum inhibitory concentrations (MICs) of six antibiotic and antiseptic compounds for all isolates (Table 1). All isolates were susceptible to vancomycin (MIC = 0.5–2 μg/mL), and all were inhibited by low concentrations of benzalkonium chloride and povidone-iodine. Larger susceptibility differences between isolates were observed for amikacin, ceftazidime, moxifloxacin, and ofloxacin. The genomes of the three isolates with the highest amikacin MICs (E374, E474, and E547) encoded aminoglycoside resistance genes (Fig 1). The same three isolates also showed higher MICs to both moxifloxacin and ofloxacin (Table 1). Enterococci are known to develop fluoroquinolone resistance via mutations in the genes encoding DNA gyrase subunit A (*gyrA*) and topoisomerase IV (*parC*) [2]. We examined the *gyrA* and *parC* sequences of the three fluoroquinolone-resistant isolates, and found that all three isolates encoded mutations at amino acid position 84 in *gyrA* and position 82 in *parC* (Table 2). The S84R and S84I mutations in *gyrA* and S82I mutation in *parC* have been

**Table 1. Minimum Inhibitory concentration (MIC) profiles of *E. faecalis* endophthalmitis isolates.**

| Strain ID | ST | MIC (µg/mL) | | | | | | |
|-----------|-----|------|-----|--------|-------|-----|-----------|-----|
| | | AMK | BKC | CZA | MOX | OFX | PI[1] | VAN |
| E374 | 2 | 512 | 2 | 312.5 | 16 | 32 | 0.31 | 1 |
| E547 | 2 | 512 | 2 | 625 | 16 | 32 | 0.31 | 1 |
| E266 | 25 | 128 | 2 | 156.25 | 0.25 | 4 | 0.31 | 0.5 |
| E687 | 34 | 64 | 2 | 312.5 | 0.125 | 2 | 0.31 | 2 |
| E77 | 40 | 128 | 2 | 312.5 | 0.25 | 2 | 0.31 | 2 |
| E139 | 40 | 128 | 1 | 625 | 0.5 | 4 | 0.31 | 2 |
| E286 | 40 | 64 | 2 | 625 | 0.5 | 2 | 0.31 | 2 |
| E263 | 55 | 128 | 2 | 78.125 | 0.25 | 2 | 0.31 | 2 |
| E46 | 64 | 64 | 2 | 312.5 | 0.25 | 2 | 0.31 | 1 |
| E171 | 64 | 256 | 2 | 625 | 0.25 | 2 | 0.31 | 1 |
| E474 | 103 | 512 | 2 | 312.5 | 32 | 32 | 0.31 | 1 |
| E616 | 122 | 128 | 2 | 78.125 | 1 | 2 | 0.31 | 1 |
| E623 | 122 | 128 | 2 | 78.125 | 1 | 4 | 0.31 | 1 |
| E633 | 122 | 128 | 2 | 625 | 1 | 2 | 0.31 | 1 |
| E676 | 428 | 128 | 2 | 1250 | 0.25 | 4 | 0.31 | 1 |

AMK = Amikacin, BKC = Benzalkonium Chloride, CZA = Ceftazidime, MOX = Moxifloxacin, OFX = Ofloxacin, PI = Povidone-iodine, VAN = Vancomycin
[1]Povidone-iodine MIC is reported as %.

described before [23, 41], and likely explain the fluoroquinolone resistance observed in these isolates. Finally, ceftazidime resistance was quite variable between isolates, with MICs ranging from 78.125–625µg/mL (Table 1). In general, isolates belonging to the same ST had similar ceftazidime susceptibilities, with MICs falling within two-fold of one another. The one exception was the three ST122 isolates, which were all isolated from the same patient [13]. The first two isolates from the patient (E616 and E623) had ceftazidime MICs of 78.125µg/mL, while the final isolate (E633) had an eight-fold higher MIC (MIC = 625µg/mL).

## Emergence of reduced ceftazidime susceptibility in a hypermutator strain isolated from recurrent endophthalmitis

Whole-genome sequencing revealed that three *E. faecalis* isolates collected from a single patient with recurrent endophthalmitis were closely related to one another [13] (Fig 1). Using the first isolate from the patient (E616) as a reference, we looked for variants that arose in E623 (isolated 16 weeks after E616) and E633 (isolated 20 weeks after E616 and then grown in thioglycolate broth for three additional months before sequencing) [13]. We identified four

**Table 2. Amino acid substitutions in *gyrA* and *parC* identified in fluoroquinolone-resistant *E. faecalis* endophthalmitis isolates.**

| Isolate ID | ST | MXF MIC (µg/mL) | Genotype | |
|-----------|--------|-----------------|-----------|-----------|
| | | | *gyrA* | *parC* |
| E286 | ST40 | 0.5 | Wild type | Wild type |
| E374 | ST2 | 16 | S84R | S82I |
| E547 | ST2 | 16 | S84R | S82I |
| E474 | ST103 | 32 | S84I | S82I |

Moxifloxacin (MXF); Minimum inhibitory concentration (MIC)

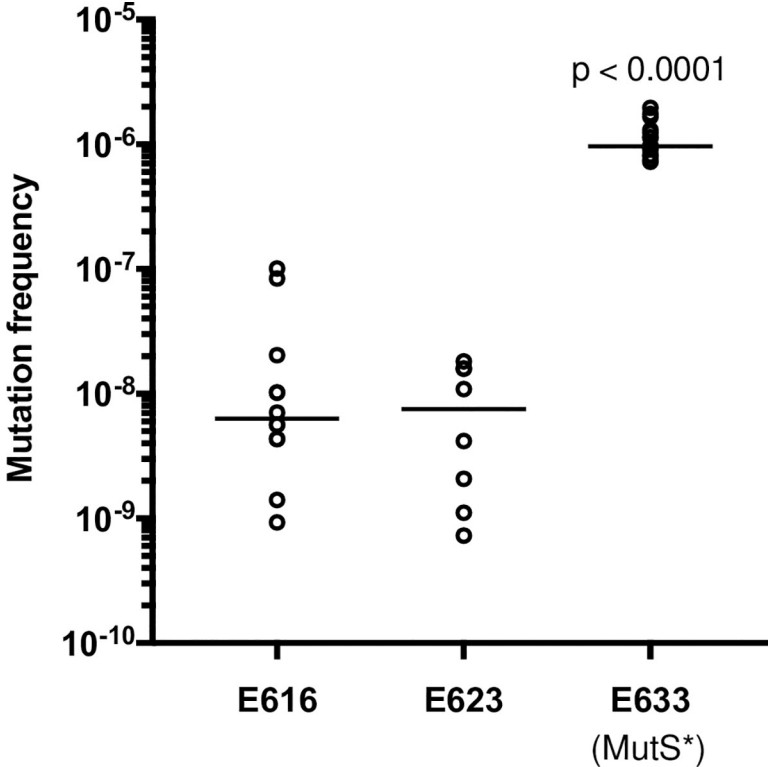

**Fig 3. An isolate from recurrent endophthalmitis with a mutation in the mismatch repair gene *mutS* has a higher spontaneous mutation rate compared to earlier isolates from the same patient.** Spontaneous mutation rate, measured with a rifampin mutagenicity assay, in isolates E616, E623 and E633, which were all sampled from a patient with recurrent endophthalmitis. Horizontal lines mark median values for results from 6 biological replicate experiments (each with 2–4 technical replicates).

variants in the E623 isolate genome compared to E616, including three small insertions and a non-synonymous mutation in a predicted arsenic resistance protein (S7 Table). The E633 isolate genome, in contrast, had 46 variants compared to E616 (S7 Table). Four of these were the same variants identified in E623, suggesting that E633 derived directly from E623. One of the new variants in E633 was a single base deletion in the DNA mismatch repair gene *mutS*, which was predicted to cause a frame-shift mutation at amino acid position 179 (S7 Table).

Mutations in *mutS* and *mutL* have been previously correlated with an increase in the rate of spontaneous mutation in *E. faecalis* [23]. To test whether the *mutS* mutation we observed was similarly associated with increased mutability, we quantified the rate of spontaneous mutation in E616, E623, and E633 using a rifampin mutagenicity assay (Fig 3). This assay has been used previously to quantify the spontaneous mutation rate in enterococci [23, 42]. The median spontaneous mutation rates of E616 and E623 were $6.3 \times 10^{-9}$ and $7.5 \times 10^{-9}$ respectively, while the mutation rate of E633 was over 100-fold higher at $1.3 \times 10^{-6}$ (Fig 3, $P<0.0001$). The increased mutation rate in E633 likely explains the excess of variants identified in this isolate, and may have also contributed to its decreased ceftazidime susceptibility and its ability to persist and cause a recurrent infection. Analysis of the variants accumulated by E633 versus E616 revealed that E633 carried mutations in genes encoding: (i) cell wall-associated proteins, such as a RodA-like rod-shape determining protein and a WxL domain-containing hypothetical protein, (ii) transcriptional regulators such as a LuxR two-component transcriptional response regulator and a Spx regulatory protein, and (iii) membrane-associated proteins such a putative

serine/threonine exporter and a phosphotransferase system sugar transporter (S7 Table). Any of these, or additional variants identified in the E633 genome, might have contributed to the decreased cephalosporin susceptibility and persistence of this isolate *in vivo*.

## Discussion

In this study, we characterized the genetic diversity and variability of antimicrobial susceptibility and virulence phenotypes among *E. faecalis* isolated from post-operative endophthalmitis. We found isolates belonging to well-known genetic lineages associated with clinical infections, as well as isolates resembling commensal enterococci lacking mobile genetic elements and other pathogenic traits. The presence of genetically diverse *E. faecalis* from endophthalmitis is in agreement with previous studies showing the absence of specific lineages that colonize the ocular environment [43]. The sequence types found in multiple endophthalmitis patients, however, included ST2, ST40, and ST64, which have all been previously found to cause infections [25]. A similarly diverse array of genetic lineages, with a subset being found more frequently, has also been observed in other Gram-positive bacteria isolated from ocular infections [44].

We found a high degree of variability in mobile genetic element content in the genomes of the isolates that we studied. The variable antimicrobial resistance genes we identified, including *erm*(B), *tet(M)*, *tet(S)*, *ant(6)-Ia*, *aph(3′)-III*, and *aac(6′)-aph(2″)*, are among the most commonly observed resistance genes in *E. faecalis* [45–47]. These resistance genes are often plasmid-encoded [48], and indeed nearly all genomes encoding acquired resistance genes also encoded one or more plasmid *rep* genes. Because we did not fully resolve the plasmid sequences in this study, however, we cannot formally connect resistance genes to the mobile elements carrying them. Cytolysin operon presence, CRISPR-Cas loci, and prophages were also variable among the *E. faecalis* we isolated. Similar to prior studies [49, 50], we detected more acquired antimicrobial resistance genes in the genomes of isolates lacking a functional CRISPR-Cas locus. The difference was not statistically significant, however, likely due to the limited number of genomes we sampled.

Virulence factors previously shown to contribute to enterococcal pathogenicity during endophthalmitis include the *E. faecalis* cytolysin [51], the fsr quorum-sensing system [52], and secreted proteases including gelatinase (GelE) and serine protease (SprE) [53]. Cytolysin operons were detected in only five of the 17 genomes analyzed here, while fsr quorum-sensing genes were detected in 13 genomes and GelE/SprE were detected in 16 genomes. Of the other virulence factors identified, some were found in all isolates while others were variably present, and when virulence genes were identified they often showed differences in nucleotide sequence compared to one another. Detailed clinical information about the patients these isolates were collected from was not available, thus we were unable to assess the relative severity of the infections. Examining correlations between virulence factors and clinical outcomes will be a focus of our future work.

The ability to form biofilms allows bacteria to persist at anatomical sites where they might normally be cleared. Biofilm formation on intraocular lenses has been suggested to contribute to bacterial endophthalmitis [54], and has been hypothesized to play a role in persistent and recurrent ocular infections [13, 55]. When we measured *in vitro* biofilm formation in the endophthalmitis isolates we collected, we observed differences across isolates, including isolates belonging to the same ST. Because same-ST isolates are often more closely related than isolates belonging to different STs, we examined the ST40 and ST64 isolate genomes for genetic differences that could account for increased biofilm formation, and focused our analysis on previously described biofilm and attachment factors. Disruption of the collagen adhesin

gene *ace* was associated with diminished biofilm formation in both ST40 and ST64 isolates. Ace mediates *E. faecalis* adherence [56], and contributes to pathogenicity in animal models of infective endocarditis and urinary tract infection [57, 58]. We also identified an Asa373-like aggregation factor gene that was only present in the ST64 isolate that made more biofilm, and which is predicted to encode a surface adhesin that contributes to *E. faecalis* aggregation [40]. Further experiments such as targeted gene disruption or complementation would be required to validate the role of either one of these genes in biofilm formation; this will be a focus of our future work in this area.

Bacterial endophthalmitis is frequently managed by pre- and post-operative administration of topical or intravitreal antibiotics, often before identifying and testing the susceptibility of the organism [59, 60]. We observed differences in bacterial susceptibility to antibiotics between the isolates we tested, and in most cases, we were able to correlate increased MICs to resistance genes or resistance-associated mutations. All isolates were vancomycin susceptible, and all lacked Van operons. Aminoglycoside resistance genes were found in the isolates with the highest amikacin MICs, and previously described *gyrA* and *parC* mutations were found in the three fluoroquinolone-resistant isolates. While we observed variability in ceftazidime MICs between isolates, it is likely that multiple genes and/or mutations contribute to the intrinsic cephalosporin resistance that is common among enterococci. This was also apparent when considering the mutations present in the hypermutator isolate E633, which was less susceptible to ceftazidime compared to closely related isolates from the same patient collected at earlier infection time points. Although the E633 isolate genome encoded mutations in cell wall proteins, membrane proteins and transcriptional regulators, they were largely unrelated to known mechanisms of cephalosporin resistance such as the CroRS signal transduction pathway [61], the serine/threonine kinase IreK [62], and MurAA [63]. Overall, from these data we can conclude that vancomycin remains an appropriate treatment for suspected enterococcal endophthalmitis, however the availability of limited antibiotics for infection control and the use of aminoglycosides, cephalosporins, and fluoroquinolones could be ineffective due to pre-existing resistance in the infecting strain.

This study had several limitations. First, our sample size was quite small, as *E. faecalis* endophthalmitis is a relatively rare occurrence. In addition, the isolates we characterized were from a convenient sampling of strains available at the UPMC Eye Center, and detailed clinical information was not available for the patients that provided these isolates. Furthermore, our genomic analysis of virulence factors and correlations with biofilm production were restricted to the STs that were sampled from multiple patients and that showed within-ST differences in biofilm formation. Finally, we are unable to discern whether the *mutS* mutation we detected in the E633 isolate arose during recurrent endophthalmitis within the patient, or whether it was selected during the extended period of incubation after isolation. Further investigation is also needed to elucidate the mechanistic connection between genetic changes and increased ceftazidime resistance in the E633 isolate.

In summary, this study shows that *E. faecalis* causing post-operative endophthalmitis are genetically diverse and phenotypically variable. Our functional genomics analysis identified expected associations between antimicrobial resistance and biofilm formation genes, as well as the unexpected occurrence of a hypermutator isolate from a recurrent infection. Further work is needed to understand the precise contributions of enterococcal virulence factors such as Ace and Asa-373 to enterococcal growth in the collagen-rich vitreous humor, and to further elucidate the role of hypermutators in ocular infections. Such studies could lead to improved treatment strategies for intra-ocular infections, which would be welcome additions to an antimicrobial treatment arsenal that is under threat from increasing antibiotic resistance.

## Supporting information

**S1 Fig. Core genome phylogeny of 17 *E. faecalis* endophthalmitis isolates plus 49 diverse *E. faecalis* from the NCBI database.** A core genome alignment was generated for 1,710 core genes with Roary, and the phylogeny was made with RAxML. Tips are labeled with isolate name, source, sequence type (ST), and year of isolation. Isolates from endophthalmitis are labeled red. Bootstrap values >90 are shown on the tree.
(TIF)

**S2 Fig. Biofilm staining versus stationary phase cell density of 15 *E. faecalis* endophthalmitis isolates.** Crystal violet OD550 values from a standard in vitro biofilm assay were compared with stationary phase OD600 values for the same isolates grown overnight in Brain Heart Infusion media supplemented with 0.25% glucose. The mean value for each isolate is plotted, and error bars indicate the standard error of the mean of 24 measurements for biofilm staining and 9 measurements for stationary phase cell density.
(TIF)

**S3 Fig. Disruption of the ace collagen adhesin in isolates with reduced biofilm formation.** Nucleotide sequence alignment of the ace sequences from (A) ST40: E286, E77 and E139, and (B) ST64: E171 and E46 isolates. Yellow arrows show the ace coding sequence, black bars show nucleotide regions that align with one another, and black lines show deletions in isolates that form less biofilm.
(TIF)

**S1 Table. Isolate information and UPMC genome assembly statistics.**
(XLSX)

**S2 Table. Publicly available *E. faecalis* isolates used for phylogenetic comparisons.**
(XLSX)

**S3 Table. CRISPR-cas loci in post-operative endophthalmitis *E. faecalis* genomes.**
(XLSX)

**S4 Table. Putative intact and questionable prophage encoding regions predicted by PHASTER.**
(XLSX)

**S5 Table. Enterococcal virulence factors present in *E. faecalis* endophthalmitis genomes.**
(XLSX)

**S6 Table. Variable gene presence/absence in ST40 and ST64 isolate genomes.**
(XLSX)

**S7 Table. Variants in recurrent endophthalmitis isolates (E623, E633) versus the earliest isolate from the same patient (E616).**
(XLSX)

## Acknowledgments

We gratefully acknowledge all members of the Van Tyne lab, and in particular Vatsala Srinivasa for assistance constructing the phylogeny shown in S1 Fig.

## Author Contributions

**Conceptualization:** Gayatri Shankar Chilambi, Robert M. Q. Shanks, Daria Van Tyne.

**Data curation:** Regis P. Kowalski, Daria Van Tyne.

**Formal analysis:** Gayatri Shankar Chilambi, Daniel R. Evans, Daria Van Tyne.

**Funding acquisition:** Daria Van Tyne.

**Investigation:** Gayatri Shankar Chilambi, Hayley R. Nordstrom, Daniel R. Evans, Regis P. Kowalski, Deepinder K. Dhaliwal, Vishal Jhanji, Robert M. Q. Shanks, Daria Van Tyne.

**Methodology:** Regis P. Kowalski, Daria Van Tyne.

**Project administration:** Daria Van Tyne.

**Supervision:** Deepinder K. Dhaliwal, Robert M. Q. Shanks, Daria Van Tyne.

**Validation:** Daria Van Tyne.

**Visualization:** Gayatri Shankar Chilambi, Hayley R. Nordstrom, Daria Van Tyne.

**Writing – original draft:** Gayatri Shankar Chilambi, Hayley R. Nordstrom, Daria Van Tyne.

**Writing – review & editing:** Gayatri Shankar Chilambi, Hayley R. Nordstrom, Daniel R. Evans, Regis P. Kowalski, Deepinder K. Dhaliwal, Vishal Jhanji, Robert M. Q. Shanks, Daria Van Tyne.

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
