## [Decision Letter · Decision Letter 0]

10 Feb 2021

PONE-D-20-40363

Genomic and phenotypic diversity of Enterococcus faecalis isolated from endophthalmitis

PLOS ONE

Dear Dr. Van Tyne,

Thank you for submitting your manuscript to PLOS ONE. After careful consideration, we feel that it has merit but does not fully meet PLOS ONE’s publication criteria as it currently stands. Therefore, we invite you to submit a revised version of the manuscript that addresses the points raised during the review process.

Please provide information on nature of mutations leading to resistance and clinical outcomes in different cases.

We look forward to receiving your revised manuscript.

Kind regards,

Iddya Karunasagar

Academic Editor

PLOS ONE

Additional Editor Comments:

The reviewers have made some suggestions. Some additional information suggested by reviewers will strenthen the manuscript. Please provide point to point response for the reviewer comments.

Reviewers' comments:

Reviewer's Responses to Questions

**Comments to the Author**

1. Is the manuscript technically sound, and do the data support the conclusions?

Reviewer #1: Yes

Reviewer #2: Yes

2. Has the statistical analysis been performed appropriately and rigorously? 

Reviewer #1: No

Reviewer #2: Yes

3. Have the authors made all data underlying the findings in their manuscript fully available?

Reviewer #1: Yes

Reviewer #2: Yes

4. Is the manuscript presented in an intelligible fashion and written in standard English?

Reviewer #1: Yes

Reviewer #2: Yes

5. Review Comments to the Author

Reviewer #1: This paper analyzed the genetic variation and phenotypic character of Enterococcus faecalis, which was identified as the causative agent of endophthalmitis, and reported its diversity. A total of 17 strains were used for the analysis, of which 15 were used at the center to which the author belongs, and 2 were reported. Whole genome sequences were analyzed and antibiotic susceptibility tests, biofilm and cytolysin assays, and genetic mutation assays were conducted to compare gene changes and phenotypic characteristics. Several experiments have shown quite good results, but the number of individuals is small, so it seems that it is not possible to draw a definite conclusion. However, infectious endophthalmitis occurs not only in one in tens of thousands of people, but also E. faecalis is less likely to occur as a causative agent. Recognizing these limitations, this paper can be seen as helpful in providing motives for various researches.

1. For the antibiotic resistance of fluoroquinolones, you have checked the mutation of gyrA. For parC, please describe how the mutation is. In fact, the reality is that there are few antibiotics available in eye clinic for infection control. Therefore, you may be more interested in antibiotic resistance to the first to fourth generations of fluoroquinolones, vancomycin, ceftazidime, cefuroxime, than other antibiotics. I hope that further research will be done on this.

2. E. faecalis, which is derived from endophthalmitis, can be very rare, but is relatively common in other diseases. Regarding this background, I think that comparing E. faecalis cultivated in other diseases can supplement the limitations of the paper.

3. Recently, there are papers that E. faecalis is increasing as a causative isolate of postoperative endophthalmitis. There are reports of these in Sweden, Taiwan, and Korea, and it would be nice to add them to the introduction.

Reviewer #2: This is an interesting, well-written, well-presented study of enterococcus samples that caused endophthalmitis at single institution. Known E. faecalis virulence factors (cytolysin and biofilm production) and antibiotic resistance genes in each genome were analyzed. Interestingly, the authors found evidence of E. faecalis adaptation during recurrent endophthalmitis by identifying genetic variants that arose in sequential isolates sampled at different time points from the same patient, however, it is not clear if the mutation arose during the recurrent endophthalmitis within the patient or if it was selected during the extended period of

incubation after isolation. All of the samples were susceptible to vancomycin.

While the variability in the virulence factors is interesting, it would be especially interesting to see if there was any clinical correlation between these virulence factors and outcomes.

6. PLOS authors have the option to publish the peer review history of their article (what does this mean?). If published, this will include your full peer review and any attached files.

Reviewer #1: No

Reviewer #2: No

---

## [Author Response · Author response to Decision Letter 0]

2 Mar 2021

Point-by-point responses to the reviewer comments are included in the Response to Reviewers document. Other journal requirements have been addressed in the updated submission.

---

## [Decision Letter · Decision Letter 1]

31 Mar 2021

Genomic and phenotypic diversity of Enterococcus faecalis isolated from endophthalmitis

PONE-D-20-40363R1

Dear Dr. Van Tyne,

We’re pleased to inform you that your manuscript has been judged scientifically suitable for publication and will be formally accepted for publication once it meets all outstanding technical requirements.

Kind regards,

Iddya Karunasagar

Academic Editor

PLOS ONE

Additional Editor Comments (optional):

All reviewer comments have been addressed satisfactorily

Reviewers' comments:

Reviewer's Responses to Questions

**Comments to the Author**

1. If the authors have adequately addressed your comments raised in a previous round of review and you feel that this manuscript is now acceptable for publication, you may indicate that here to bypass the “Comments to the Author” section, enter your conflict of interest statement in the “Confidential to Editor” section, and submit your "Accept" recommendation.

Reviewer #1: All comments have been addressed

Reviewer #2: All comments have been addressed

2. Is the manuscript technically sound, and do the data support the conclusions?

Reviewer #1: Yes

Reviewer #2: Yes

3. Has the statistical analysis been performed appropriately and rigorously? 

Reviewer #1: Yes

Reviewer #2: Yes

4. Have the authors made all data underlying the findings in their manuscript fully available?

Reviewer #1: Yes

Reviewer #2: Yes

5. Is the manuscript presented in an intelligible fashion and written in standard English?

Reviewer #1: Yes

Reviewer #2: Yes

6. Review Comments to the Author

Reviewer #1: They answered the reviewer's question without any special problems. I think that it is practically inevitable to have a small number of E. faecalis bacteria isolated from postoperative endophthalmitis. I wonder if there is a possibility that the shortage of strains will be solved if international cooperative research is carried out.

Reviewer #2: The authors have addressed the comments. They are unable to make clinical correlations but identify it as a focus of future work.

7. PLOS authors have the option to publish the peer review history of their article (what does this mean?). If published, this will include your full peer review and any attached files.

Reviewer #1: **Yes: **Sang Joon Lee

Reviewer #2: No

---

## [Editor Report · Acceptance letter]

5 Apr 2021

PONE-D-20-40363R1 

Genomic and phenotypic diversity of *Enterococcus faecalis* isolated from endophthalmitis 

Dear Dr. Van Tyne:

I'm pleased to inform you that your manuscript has been deemed suitable for publication in PLOS ONE. Congratulations! Your manuscript is now with our production department. 

Kind regards, 

on behalf of

Dr. Iddya Karunasagar 

Academic Editor

PLOS ONE